# The Habitat Type and Scale Dependences of Interspecific Associations in a Subtropical Evergreen Broad-Leaved Forest

Changchun Jiang [1,†], Jiaqin Fu [1,†], Yunquan Wang [1,*], Pengtao Chai [1], Yidan Yang [1], Xiangcheng Mi [2], Mingjian Yu [3], Keping Ma [2] and Jianhua Chen [1]

[1] College of Chemistry and Life Sciences, Zhejiang Normal University, Jinhua 321004, China
[2] State Key Laboratory of Vegetation and Environmental Change, Institute of Botany, Chinese Academy of Sciences, Beijing 100093, China
[3] Ministry of Education Key Laboratory of Conservation Biology for Endangered Wildlife, College of Life Sciences, Zhejiang University, Hangzhou 310058, China
[*] Correspondence: yqwang@vip.126.com
[†] These authors contributed equally to this work.

**Abstract:** "Interspecific associations" refers to the interrelationship among different species in a particular spatial distribution, which plays an important role in species distribution, community assembly, and responses to environmental changes. However, the strength and/or direction of interspecific associations may vary with environmental gradients and scales. To understand the effects of habitat types and research scales on interspecific associations in subtropical forests, we modeled the interspecific associations for more than 15,000 individuals representing 74 co-occurring species from three habitat types and three scales by using the variance ratio and the Spearman rank correlation coefficient. We found that overall interspecific associations at a community level exhibited significant positive associations for most habitat types and scales. Moreover, interspecific associations of pairwise species have strong habitat dependence, and the association strengths decreased with the increase in elevation (change in habitat types). However, the scale dependence of pairwise interspecific associations varies with habitat types. The strength of interspecific associations increased with the increasing scales (10 m × 10 m, 20 m × 20 m, and 40 m × 40 m) at low-valleys and mid-hillside habitats, while the scale-dependent effect was not detected at high-ridges. In conclusion, our study highlights the importance of environmental gradients and research scales on interspecific associations in diverse subtropical forests, and environmental gradients and research scales should be considered in future studies.

**Keywords:** interspecific associations; habitat dependences; scale dependences; interspecific correlations; subtropical forests

## 1. Introduction

"Interspecific association" refers to the interactions and spatial relationships among different species occupying a habitat, which can provide prominent insights into community assembly as well as determine community structure and function [1–4]. This concept is widely used to quantify interspecific relationships and to infer ecological processes underlying community assembly [5–9]. Previous studies have shown that the interspecific associations of plant communities are not static and are usually constrained by a variety of influencing factors such as interspecific competition [10], environmental changes [11,12], and research scales [8]. However, little is known about how the strengths and/or directions of interspecific associations vary with changing environmental gradients and research scales.

Environmental factors are important in that they affect the interspecific associations of communities, among which factors such as elevation, convexity, and slope have important influences on the spatial pattern and species richness of communities [13–15]. For example,

Chesson [16] suggested that the status of dominant species may be more affected by habitat changes than by other factors in multispecies-competing communities. Zhang et al. [17] found significant correlations between vegetation patterns and habitat factors such as elevation and slope in the Taihang Mountains in China. Moreover, Long and Tang suggested that the effects of topographic factors (such as elevation and slope) on spatial structure vary with the changing topographic factors in an evergreen broad-leaved forest [18]. These findings suggest that the strengths and directions of interspecific associations change along environmental gradients.

Research scale is also a critical factor affecting the strengths of the interspecific associations of communities [8,19]. Previous studies have shown that research scales should be matched to the selected plot area, plant community type, and vegetation uniformity when detecting interspecific associations [20]. For instance, Deng found that the overall interspecific associations at community level changed from negative to positive correlation accordingly when the research scales changed from the small scales of 100 m$^2$ and 200 m$^2$ to the large scale of 400 m$^2$ in a mixed coniferous and broad-leaved forest [21]. Deng et al. found that with the gradual increase in the scale, the percentage of significant positive interspecific associations of a secondary forest had a clear trend of first increasing and then slowly decreasing [22]. Thus, research scales should be considered when quantifying the interspecific associations of communities.

The subtropical evergreen broad-leaved forest is a typical zonal vegetation type in China. The richness of the vegetation diversity of the subtropical evergreen broad-leaved forest is second only to that of the tropical rainforest and much higher than that of other evergreen broad-leaved forests [23–25]. The interspecific interactions and biodiversity maintenance mechanisms in subtropical forests have been core questions for ecologists in China [24,26,27]. This provides an ideal site for studying how interspecific associations vary with changes in habitat type and research scale.

To evaluate the effects of habitat type and research scale on interspecific associations in subtropical forests, we examined how the strength and/or direction of interspecific associations for more than 15,000 individuals representing 74 co-occurring species change with habitat type and research scale in a subtropical forest plot in China. Specifically, we focused on the following questions:

(1) How do the overall interspecific associations at community level vary with habitat type and research scale in the subtropical forest?

(2) What is the habitat dependence of the pairwise interspecific associations in the subtropical forest?

(3) What is the scale dependence of the pairwise interspecific associations in the subtropical forest?

## 2. Materials and Methods

### 2.1. Study Site and Tree Census

The study was carried out in the predominantly old-growth subtropical forest of the 5-ha forest dynamic plot (FDP; 29.25° N, 118.12° E) in the Qianjiangyuan National Park, China. This region is located in the mid-subtropical humid monsoon climate zone in China, which has rich rainfall in summer, warm and humid weather in winter, and abundant microclimate habitats [28]. The mean annual temperature in this region is 15.3 °C, and the mean annual precipitation is 1964 mm [12]. The main soil types in this region are red soil, yellow-red soil, red-yellow soil, and, in some areas, swamp soil [28]. Qianjiangyuan National Park is rich in forest vegetation types, with evergreen broad-leaved forests, evergreen and deciduous broad-leaved mixed forests, coniferous broad-leaved mixed forests, coniferous forests, and subalpine wetland [29]. Among these, the mid-subtropical evergreen broad-leaved forest is the main vegetation in the park, a typical mid-subtropical zonal vegetation that is widely distributed in low-elevation areas below 800 m above sea level.

The 5-ha forest dynamic plot was established in a large ditch between the two main hills in accordance with CTFS-ForestGEO protocol [30]. The plot is 200 m long from east to west and 250 m wide from north to south (Figure 1). All freestanding woody stems with DBH (diameter at breast height, 1.3 m) ≥ 1 cm in the plot were tagged, mapped, and identified to species during the summer of 2002 [31]. Survivorship was assessed and new recruits were tagged, measured, mapped, and identified for each 5-year interval [12]. The 5-ha plot was dominated by evergreen tree species, accounting for 52.45 % of the total species, and the most dominant species in this plot was *Castanopsis eyrei* Champ. ex Benth. Hutch., 1905 [31].

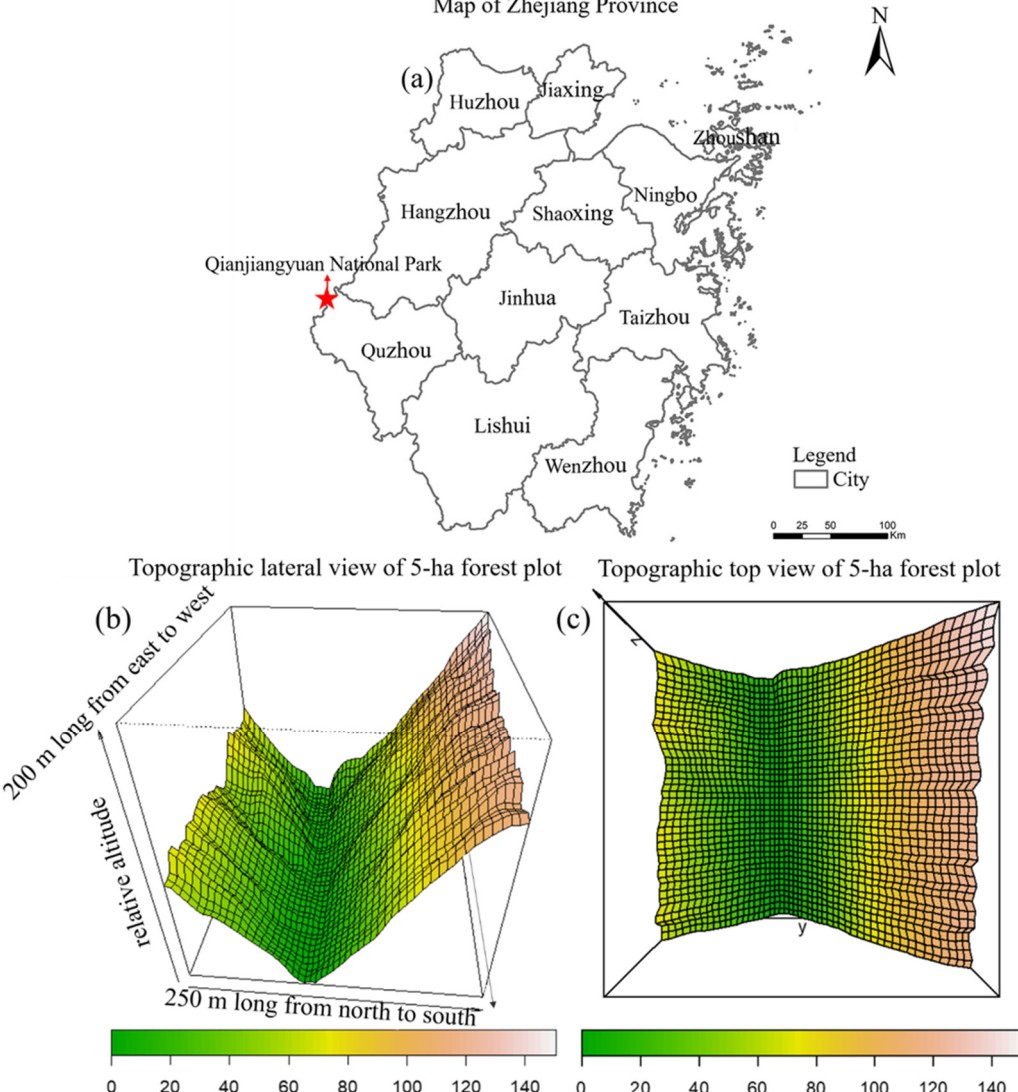

**Figure 1.** (**a**) Location of Qianjiangyuan National Park, and (**b**) lateral view and (**c**) top view of the 5-ha forest plot in Qianjiangyuan National Park.

In the present study, we used the 2012 tree census data to assess the habitat type and scale dependence of interspecific association in the 5-ha subtropical forest. In total, there were 17,398 woody plants belonging to 158 species in the 5-ha forest plot. In order to study the habitat dependence and scale dependence of the interspecific associations of the pairwise species, we analyzed all the co-occurring species in all the habitat types and scale cells. Here we modeled the interspecific associations of more than 15,000 individuals (>86% of total individuals) belonging to 74 wood species in the 5-ha subtropical forest (Table 1).

**Table 1.** Basic information about the habitat categories of the 5-ha forest plots at different scales.

| Research Scales | Habitat Categories | Habitat Types | No. of Plots | Total Area (ha) | No. of Species | No. of Co-Occurring Species | Individuals /Plot |
|---|---|---|---|---|---|---|---|
| 10 m × 10 m | H1 | Low-valleys | 200 | 2 | 146 | 74 | 36.58 |
| | H2 | Mid-hillsides | 180 | 1.8 | 134 | 74 | 29.37 |
| | H3 | High-ridges | 100 | 1 | 82 | 74 | 24.73 |
| 20 m × 20 m | H1 | Low-valleys | 50 | 2 | 146 | 74 | 146.30 |
| | H2 | Mid-hillsides | 45 | 1.8 | 134 | 74 | 117.49 |
| | H3 | High-ridges | 25 | 1 | 82 | 74 | 98.92 |
| 40 m × 40 m | H1 | Low-valleys | 11 | 1.76 | 144 | 74 | 580.36 |
| | H2 | Mid-hillsides | 10 | 1.6 | 130 | 74 | 454.50 |
| | H3 | High-ridges | 5 | 0.8 | 80 | 74 | 383.20 |

Notes: N: the total number of plots in the selected area community; H1: low-valleys; H2: mid-hillsides; H3: high-ridges.

### 2.2. Environmental Variables and Habitat Categories

In this study, we defined environmental variables in terms of topography. Four topographic factors were identified: elevation, convexity, slope, and aspect for each 20 × 20 m cell, following Harms et al. [32] and Valencia et al. [33]. To test how interspecific associations vary with habitat categories, these four topographic variables were selected by Multivariate regression tree (MRT) procedure by using *mvpart* package [34], and a tree with three habitat categories was selected as the best one after 1000 cross-validation trials following Chen et al. [35]. Based on these four variables, the whole 5-ha forest plot was classified into three habitat categories with 20 m × 20 m cells (Figure 2): low-valleys (H1, 50 plots), mid-hillsides (H2, 45 plots), and high-ridges (H3, 35 plots). Although four habitat factors—including elevation, convexity, slope, and slope direction—were used as independent variables in the analysis, the multivariate regression tree was successfully constructed only by the factor of elevation. This means that elevation played a dominant role in the habitat classification of the present study and that elevation increases with the increasing number of habitat types.

**Figure 2.** Habitat categories of the 5-ha forest plot at 20 m × 20 m scale. The white represents low-valleys (H1), the blue represents mid-hillsides (H2), and the yellow represents high-ridges (H3).

In the present study, to test how interspecific associations vary with different scales, the 5-ha forest plot was grouped into cells 10 m × 10 m, 20 m × 20 m, and 40 m × 40 m in size. There were only 8 cells 80 m × 80 m, too few for statistical analysis. Based on these three scales, the entire 5-ha forest plot was divided into different scale cells (See Figures S1 and S2 for more details). These cells were used to quantify the scale dependence of interspecific associations in the 5-ha forest plot.

### 2.3. Statistical Analysis

Community-level interspecific association: The variance ratio test (*VR*) was used to determine the overall interspecific association at community level. The *VR* value indicates whether there is a significant relationship among multiple species in the selected area, and the significance of the *VR* value was tested by the statistic *W* [1]. The *VR* value was calculated as:

$$VR = \frac{\frac{1}{N}\sum_{i=1}^{N}(T_i - t)^2}{\sum_{j=1}^{S}(1 - P_j)} \tag{1}$$

where *VR* is the variance ratio, $N$ is the total number of plots in the community, $T_i$ is the total number of target species in plot *i*. $t$ is the observed mean number of species in all the plots. $S$ represents the total number of species in the community, $P_j$ represents the frequency of species *j*, and $P_j = \frac{n_j}{N}$, $n_j$ represents the total number of plots occupied by species *j* [1].

Under the null hypothesis of independence, the expected value of *VR* is 1; that is, when *VR* = 1, it means that there is no connection between the species. If *VR* > 1, it means that there is a positive connection between the species. If *VR* < 1, it means that there is a negative connection between the species. The statistic *W* was used to verify the significant degree of *VR* deviation from 1, $W = VR \times N$. If $W > \chi_{0.05}^2(N)$ or $W < \chi_{0.95}^2(N)$, it means that the overall connection between the species is significant. Conversely, the overall connection between the species is nonsignificant.

Pairwise interspecific association: The Spearman rank correlation coefficient was used to test the interspecific association between pairwise species and quantitatively analyze the covariant linear relationship between species pairs following Bishara & Hittner [36]:

$$r(i,j) = 1 - \frac{6\sum_{p=1}^{N}(x_{ip} - \bar{x}_i)^2(x_{jp} - \bar{x}_j)^2}{N^3 - N}$$

where $r(i,j)$ represents the spearman rank correlation coefficient species *i* and species *j* in plot *p*; $N$ represents the total number of plots in the community; $x_{ip}$ and $x_{jp}$ represent the rank of species *i* and species *j* in plot *p*, respectively; and $\bar{x}_i$ and $\bar{x}_j$ represent the mean abundance of species *i* and species *j* in all the plots. The value range of $r(i,j)$ is from −1 to 1, with positive values indicating positive correlation and negative values indicating negative correlation.

All the analyses were conducted in the R 4.1.2 statistical platform (http://www.r-project.org/, accessed on 1 November 2021). Interspecific association was calculated using the function in the *spaa* package [37]. The significance of the Spearman rank correlation coefficients was calculated using the *rcorr()* function in the *Hmisc* package [38]. The visualization of matrix saliency was implemented using the *chart.Correlation()* function in the *Performance Analytics* package [39]. Boxplots were used to show the change in interspecific association across habitat categories or scales, and the *ggplot2* package [40] and *ggsignif* package [41] were used to draw boxplots and calculate the significance between different groups.

## 3. Results

### 3.1. Overall Interspecific Associations at Community Level

In order to study the habitat type and scale dependence of overall interspecific associations at community level, we evaluated the overall interspecific association of the community in three habitats and three scales by using the variance ratio method (Table 2, 9 groups in total). The overall interspecific association at community level for most habitat types and scales exhibited positive associations ($VR > 1$) while showing a negative association in habitat H3 at a scale of 20 m $\times$ 20 m ($VR = 0.846 < 1$). The $W$ statistics and $\chi^2$ tests were employed to test the significance of the deviation of the $VR$ values from 1. The overall interspecific associations at community level showed significant associations for most of the habitat types at different scales (Table 2, $W > \chi^2_{0.05}(N)$ or $W < \chi^2_{0.95}(N)$). However, overall community-interspecific associations were not significant for habitat H3 at either 20 m $\times$ 20 m or 40 m $\times$ 40 m scales.

**Table 2.** The overall interspecific associations at community level of three scales and habitats.

| Research Scales | Habitat Categories | Variance Ratio (*VR*) | *W* Statistic | $\chi^2_{0.95}(N)$, $\chi^2_{0.05}(N)$ | Test Results |
|---|---|---|---|---|---|
| 10 m $\times$ 10 m | H1 | 2.278 | 410.006 | 149.969, 212.304 | significant positive correlation |
| | H2 | 5.017 | 993.402 | 166.444, 231.829 | significant positive correlation |
| | H3 | 1.478 | 147.804 | 77.929, 124.324 | significant positive correlation |
| 20 m $\times$ 20 m | H1 | 2.506 | 112.757 | 30.612, 61.656 | significant positive correlation |
| | H2 | 6.378 | 318.903 | 34.764, 67.505 | significant positive correlation |
| | H3 | 0.846 | 21.139 | 14.611, 37.652 | Nonsignificant negative correlation |
| 40 m $\times$ 40 m | H1 | 2.644 | 26.439 | 3.940, 18.307 | significant positive correlation |
| | H2 | 4.797 | 52.772 | 4.575, 19.675 | significant positive correlation |
| | H3 | 1.148 | 5.741 | 1.145, 11.070 | Nonsignificant positive correlation |

Note: N: the total number of plots in the selected area community; H1: low-valleys; H2: mid-hillsides; H3: high-ridges.

### 3.2. Habitat Dependence of Interspecific Associations for Pairwise Species

To understand the habitat type–dependence of interspecific associations for pairwise species in this forest community, we calculated the pairwise species–interspecific associations of 74 co-occurring species from three habitat types at three scales by using the Spearman rank correlation coefficient (Figure 3). The interspecific association of the pairwise species exhibited a positive association at three scales for two habitat types (Figure 3, H1 and H2), while the interspecific associations were negative (tended to be non-associated) in habitat H3 at a scale of 10 m $\times$ 10 m and 40 m $\times$ 40 m (Figure 3). The strength of the pairwise interspecific associations decreased with the increase in elevation (from H1 to H3). Moreover, the strength of the pairwise interspecific associations varied significantly among most of the habitat types at three scales, indicating that habitat type significantly affects pairwise interspecific associations in this forest. However, there were no significant associations between habitat H2 and H3 at a 40 m $\times$ 40 m scale (Figure 3c).

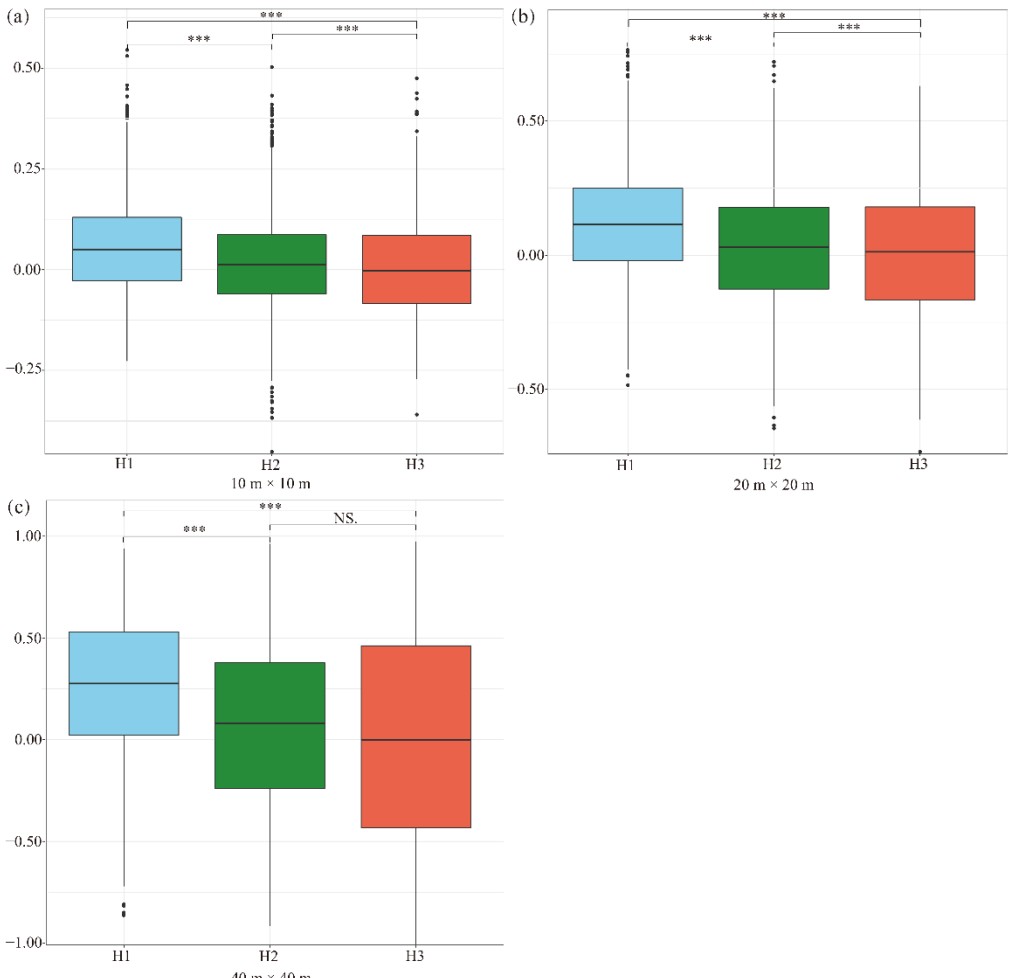

**Figure 3.** Boxplots of interspecific associations of three habitats at different scales: 10 m × 10 m scale (**a**), 20 m × 20 m scale (**b**), and 40 m × 40 m scale (**c**). Asterisks indicate a significant difference in different habitat types (*** $p$ < 0.001; NS, nonsignificant).

### 3.3. Scale Dependence of Interspecific Associations for Pairwise Species

To explore the scale dependence of interspecific associations for pairwise species in the forest community, we evaluated the pairwise species–interspecific associations of 74 co-occurrence species at 3 scale levels for each of 3 habitats types (Figure 4). The interspecific association of the pairwise species for most of the scales exhibited positive associations in all the habitat types (Figure 4). However, the interspecific associations were negative (tended to be non-associated) at a scale of 10 m × 10 m and 40 m × 40 m in habitat H3. The strength of the pairwise interspecific associations increased with the increase in research scales in habitat H1 and H2 (Figure 4). In addition, the strength of the pairwise interspecific associations changed significantly among all the scales in habitat H1 and H2 (Figure 4a,b). For habitat H3, however, a significant difference was detected only between the scales of 10 m × 10 m and 40 m × 40 m (Figure 4c). These results indicated that there were significant scale dependences in the subtropical forest in Qianjiangyuan National Park, yet the scale dependence of the pairwise interspecific associations varied with habitat types.

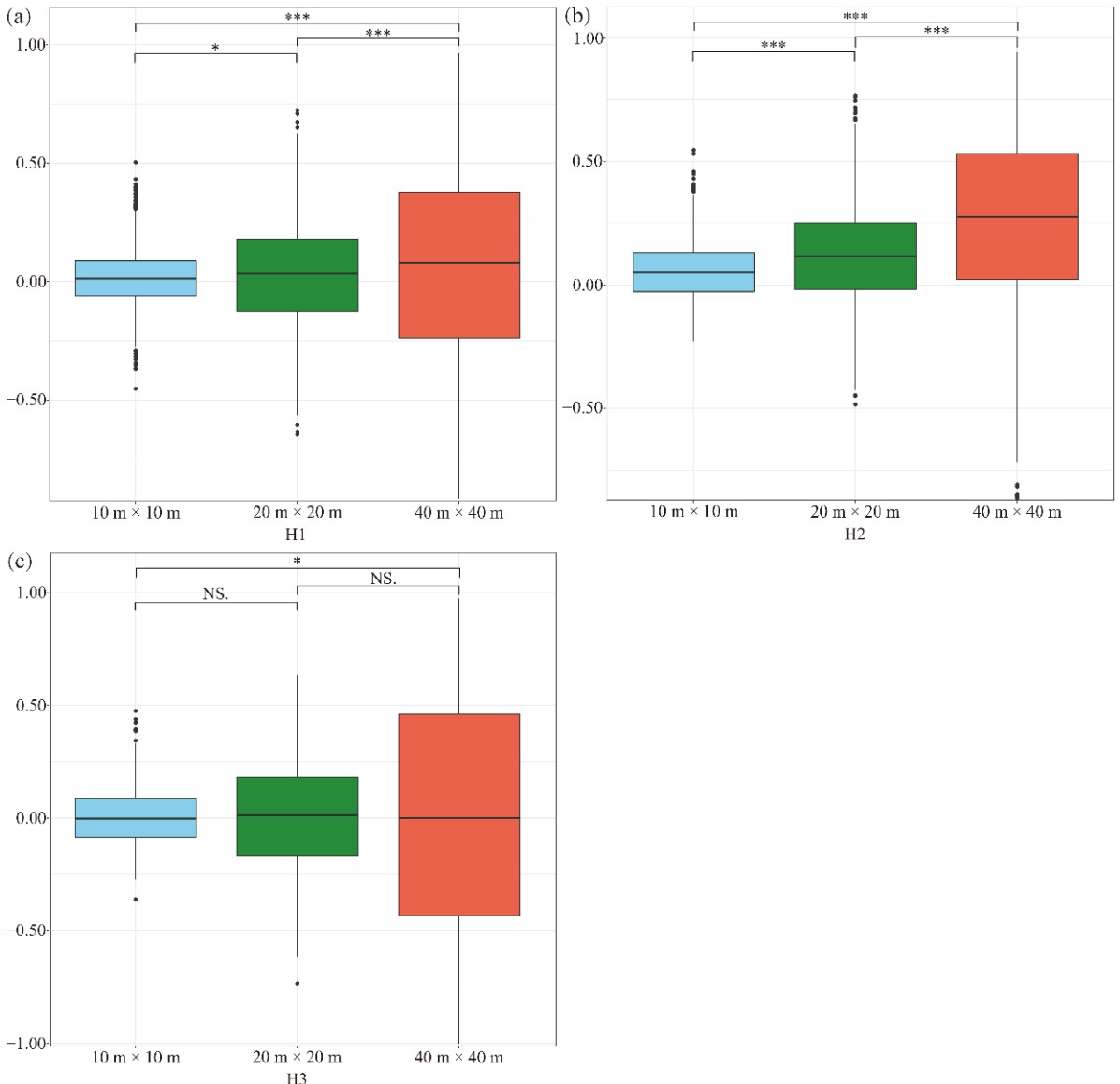

**Figure 4.** Boxplots of interspecific associations of plots at three scales from different habitat types: low-valleys (**a**), mid-hillsides, (**b**) and high-ridges (**c**). Asterisks indicate a significant difference in different habitat types (* $p < 0.05$; *** $p < 0.001$; NS, nonsignificant).

## 4. Discussions

Interspecific associations are known to influence species distribution, community assembly, and responses to environmental changes [4,8,42,43]. The importance of interspecific associations on the forest community has been studied at length, but how interspecific associations depend on environmental gradients and research scales and the nature of their interactions remain unclear. In this study, we revealed how interspecific associations depend on environmental gradients and research scales in a subtropical evergreen broad-leaved forest by using the variance ratio and the Spearman rank correlation coefficient.

### 4.1. Overall Interspecific Associations in the Gutianshan Subtropical Forest

The overall interspecific associations are known to reflect the interspecific competition among species and the stability of species composition and community structure in the community [9,44–46]. We found that the overall interspecific associations were significantly positive in the old-growth forest (Table 2). The results indicated that the Gutianshan forest was in a relatively stable succession stage in which community structure and species

composition tend to be complete and dynamically balanced [3,45]. Our findings highlight the importance of environmental filtering: Species in this forest tend to share similar environmental space and resources, resulting in overall positive associations and ultimately achieving stable coexistence [3,47]. Moreover, we also found negative overall interspecific associations for some specific habitats such as H3 at scale of 20 m × 20 m, but the effect was not statistically significant. This pattern could be due to the fact that species have different survival strategies in different habitats [11,48], and the final manifestation is that the interspecific associations change with environmental gradients such as elevation [9].

### 4.2. Habitat Dependences of Pairwise Interspecific Associations in the Subtropical Forest

Our analysis revealed significant habitat dependences of pairwise interspecific associations in the Gutianshan subtropical forest (Figure 3). The pairwise interspecific associations were positive in two of three habitat types (H1 and H2). The results of the pairwise positive associations we found here are consistent with the results of the overall interspecific associations we discussed above, indicating that environmental filtering plays an important role in pairwise species interactions in the Gutianshan forest [12,35,49]. Conversely, the pairwise interspecific associations tended to be negative- or non-associated in habitat H3. These results may due to the fact that, in addition to the effects of environmental filtering, interspecific competition for limited spaces and resources also plays an important role in determining the outcome of interspecific associations [6,11,50]. Thus, a negative relationship due to species competition can partly mask the positive relationship resulting from environmental filtering. The final detected strength of pairwise interspecific associations depends on the relative importance of the environmental filtering and the species competition.

More importantly, we found that the strength of pairwise interspecific associations varied significantly among most of the habitat types at all three scales, indicating strong habitat dependences of interspecific associations in the community. This is consistent with studies reporting that pairwise interspecific associations significantly changed with changes in altitude or environmental condition [9,43,48]. As habitat type, topography, and/or soil conditions changed greatly, species density also varied significantly across habitat types [51], finally resulting in a significant difference in species interactions across habitats types [35,43]. Interestingly, we found a decrease in pairwise interspecific associations as elevation increased (from H1 to H3), and this pattern is consistent across all three research scales. The results indicated that environmental filtering may play more important roles at a lower elevation habitat relative to the higher elevation habitat in the Gutianshan subtropical forest.

### 4.3. Scale Dependence of Interspecific Associations for Pairwise Species

Pairwise interspecific associations may not only vary across environmental gradients but also change along research scales, and scale is crucial when detecting and predicting interspecific associations [8,47]. It remains unclear how the strengths of pairwise interspecific associations vary from a small to a large scale [19]. The pairwise interspecific associations were positive for most scales and tend to be negative at other scales (Figure 4), consistent with the results of overall and pairwise interspecific associations we have discussed above.

The strength of pairwise interspecific associations varied significantly among all the scales in habitat H1 and H2 (Figure 4a,b), which indicated that research scale is an important driver of interspecific associations [8,47]. However, for habitat H3, the strength was significantly changed only between the scales of 10 m × 10 m and 40 m × 40 m (Figure 4c). Our results indicated that there were significant scale dependences in the Gutianshan subtropical forest, yet the scale dependences of pairwise interspecific associations vary with habitat types. Additionally, we found that the strength of positive pairwise interspecific associations increased with the increase in research scales in habitat H1 and H2. This pattern could be due to the fact that interspecific competition for limited space and resources mainly focuses on small scales, that neighborhood interactions are typically negligible beyond

20 m, and that species interactions are influenced by environmental factors beyond this scale [52].

## 5. Conclusions

The habitat and scale dependence of the interspecific associations in the subtropical forest reported here provide insight into how overall and pairwise interspecific associations vary with environmental gradients and research scales [20]. The interspecific associations were significantly affected by habitat type and research scale in the subtropical forest in Qianjiangyuan National Park. Community-level overall interspecific associations were significantly positive for most of the habitat types and scales. Pairwise interspecific associations have significant differences between habitat types, indicating a strong habitat type dependence of interspecific associations for pairwise species in this forest. Moreover, the interspecific association strengths decreased with the increase in elevation (from H1 to H3). For a scale-dependent effect, however, the scale dependence of pairwise interspecific associations varies with habitat types. The scale-dependent effects were detected at low-valleys (H1) and mid-hillside (H2) habitats, but not at high-ridges (H3). Our results suggest that environmental gradients and research scales significantly influence interspecific associations in diverse subtropical forests, and environmental gradients and research scales should be considered in future studies.

**Supplementary Materials:** The following supporting information can be downloaded at: https://www.mdpi.com/article/10.3390/f13081334/s1, Figure S1: Habitat categories of the 5-ha forest plot at 10 m × 10 m scale. The white color represents low-valleys (H1), the blue color represents mid-hillsides (H2), while the yellow color represents high-ridges (H3); Figure S2: Habitat categories of the 5-ha forest plot at 40 m × 40 m scale. The white color represents low-valleys (H1), the blue color represents mid-hillsides (H2), while the yellow color represents high-ridges (H3).

**Author Contributions:** C.J., Y.W. and J.C. conceived and designed the study; C.J. and J.F. performed statistical analyses and wrote the first draft with substantial input from Y.W. and J.C.; all the other authors provided data and contributed to the development of the final manuscript. All authors have read and agreed to the published version of the manuscript.

**Funding:** This study was financially supported by the Zhejiang Provincial Natural Science Foundation of China (LQ22C030001).

**Institutional Review Board Statement:** Not applicable.

**Informed Consent Statement:** Not applicable.

**Data Availability Statement:** Data are available from the authors on request.

**Acknowledgments:** We acknowledge the Center of Ecology and Resources, Qianjiangyuan National Park for support of this study, as well as the hard work of the hundreds of people who were involved in the collection of the vast quantity of data in the Gutianshan plot over the past few years.

**Conflicts of Interest:** The authors declare no conflict of interest.

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
