# Peer review of "The Habitat Type and Scale Dependences of Interspecific Associations in a Subtropical Evergreen Broad-Leaved Forest"

_forests, doi:10.3390/f13081334_

Round 1
Reviewer 1 Report
For methods:
I believe that a location map of the area was missing to illustrate where the study was carried out.
In the selection of variables and habitats, why did you not use soil data (physical and chemical)? An interesting piece of information would be to use the penetrometer to qualify and quantify soil compaction, since it influences the distribution of species, therefore it influences their associations.
Why didn't they use the Chi-Square test to look for associations?
Results
In figure 3 what do figures a, b and c mean? What do they represent? It's not in the caption.
Author Response
Reviewer(s)' Comments to Author:
Reviewer 1
For methods:
I believe that a location map of the area was missing to illustrate where the study was carried out.
Response: In our revised manuscript, we add a map to show the location of Gutianshan 5-ha forest plot in Figure 1.
In the selection of variables and habitats, why did you not use soil data (physical and chemical)? An interesting piece of information would be to use the penetrometer to qualify and quantify soil compaction, since it influences the distribution of species, therefore it influences their associations.
Response: Thank you for pointing out this. We agree with reviewer that soil variables have the potential to impact species associations. However, soil data was not available for the Gutianshan 5-ha forest plot, which constrains our ability to forecast the influences of soil variables on interspecific associations. In our manuscript, we classify habitat types based on topography variables and mainly focus on topography-based habitat dependence of interspecific species associations.
Why didn't they use the Chi-Square test to look for associations?
Response: Chi-Square test could be used for qualitative description and cannot well reflect the quantitative changes of the two variables. The Spearman rank correlation coefficient is calculated from the quantitative data, which can better test the quantitative changes of variables. Thus, Spearman rank correlation coefficient rather than Chi-Square test was used to test the associations in the manuscript.
Results
In figure 3 what do figures a, b and c mean? What do they represent? It's not in the caption.
Response: In figure 3, a indicates 10 m × 10 m scale, b indicates 20 m × 20 m scale, and c indicates 40 m × 40 m scale. We made meaning of a, b and c clear in the revised manuscript for both figure 3 and figure 4.

Reviewer 2 Report
The introduction is appropriate in addressing, first the definition of interspecific association, its usefulness in community assembly, structure and function and second, the environmental factors and research scales governing the strengths and directions of interspecific associations
The 15,000 individuals analyzed is a sample of the total number of individuals in the plot. It is not clear how they were sampled. Please, explain!
In the statistical analysis, they tested at the community level the interspecific association. However, it is not clear what the community is here. It needs to be clearly defined, described or characterized. Is it the whole plot?
The results are consistent and well presented with respect to the main axis of the research
In the discussion, the terms: Gutianshan subtropical forest, old-growth forest, just appear in the discussion with no connection with the previous part of the manuscript. This is not coherent and needs to be clarified
In the discussion also, you need to pay attention to grammar and sentence construction in order to make your text more understandable
In a conclusion, there is normally no citation. Furthermore, the term dramatically should be replaced by significantly
Good luck!
Author Response
Reviewer(s)' Comments to Author:
Reviewer 2
The introduction is appropriate in addressing, first the definition of interspecific association, its usefulness in community assembly, structure and function and second, the environmental factors and research scales governing the strengths and directions of interspecific associations
Response: Thank you for your detailed comments and suggestions which helped us improve the manuscript greatly.
The 15,000 individuals analyzed is a sample of the total number of individuals in the plot. It is not clear how they were sampled. Please, explain!
Response: Actually, there were 17, 398 individuals for the whole plot. In order to study the habitat dependence and scale dependence of interspecific associations of pairwise species, we analyzed all co-occurring species in all habitat type and scale cells. There were more than 15, 000 individuals belong to all co-occurring species were analyzed in the manuscript. Please find more detailed information about the explanation in L109-L115.
In the statistical analysis, they tested at the community level the interspecific association. However, it is not clear what the community is here. It needs to be clearly defined, described or characterized. Is it the whole plot?
Response: Yes, in simple terms, community level means in whole plot level in our study. We used term “Overall interspecific associations” to represent community level interspecific associations in the manuscript.
The results are consistent and well presented with respect to the main axis of the research
Response: Thanks for your recognition.
In the discussion, the terms: Gutianshan subtropical forest, old-growth forest, just appear in the discussion with no connection with the previous part of the manuscript. This is not coherent and needs to be clarified
Response: Thank you for pointing out. In the revised manuscript, we made the terms clear as best we can. Such as L91-L97.
In the discussion also, you need to pay attention to grammar and sentence construction in order to make your text more understandable
Response: we made some changes in discussion part to make the sentence more understandable in the revised manuscript.
In a conclusion, there is normally no citation. Furthermore, the term dramatically should be replaced by significantly
Response: Thank you for pointing out. We replaced “dramatically” with “significantly” in the revised manuscript.
